# The potential impacts of exploitation on the ecological roles of fish species targeted by fisheries: A multifunctional perspective

Eudriano F. S. Costa [1,2]*, Gui M. Menezes[1,2], Ana Colaço[1,2]

1 IMAR- Instituto do Mar, University of the Azores, Horta, Portugal, 2 OKEANOS- Institute of Marine Sciences, University of the Azores, Horta, Portugal

* eudrianocosta@gmail.com

## Abstract

Examining ecosystem functioning through the lens of trait diversity serves as a valuable proxy. It offers crucial insights into how exploitation affects the specific ecological roles played by fisheries targeted species. The present study investigates the potential impacts of exploitation on the ecological roles of fish species targeted by fisheries through an examination of trait diversity. It focuses on the trait diversity of fish landed by local and coastal fleets in the Azores archipelago over the past four decades. Fourteen functional traits were merged to data on fish assemblages landed by both fishing fleets from 1980 to 2020. These traits corresponded to four fundamental fish functions: habitat use, locomotion, feeding and life history. Variability in functional diversity metrics (i.e., functional richness- FRic, functional evenness- FEve, functional divergence-FDiv, and functional dispersion- FDis) among fleets, functions and across decades was assessed using null models. The results revealed similar trait diversity between assemblages landed by local and coastal fishing fleets with overall trait diversity remaining relatively stable over time. However, fishery activities targeted a wide range of functional traits. Additionally, seasonal availability and increased catches of certain fish species can significantly alter trait diversity and their associated functions. The findings highlight the importance of addressing fishing impacts on species traits and their ecological roles, which is crucial for long-term fisheries and ecological sustainability.

## 1. Introduction

Overfishing poses a significant threat to marine ecosystems worldwide, with implications for both ecological integrity and human well-being [1–4]. The depletion of fish stocks and declining fisheries yields signal a pressing need for enhanced monitoring and management measures to address this critical issue [1, 5]. Overexploitation of marine resources not only jeopardizes the sustainability of fisheries but also undermines the resilience of marine ecosystems to anthropogenic stressors and climate change [6–8]. When fish stocks are depleted beyond sustainable levels, it disrupts the delicate balance of marine food webs and ecosystem dynamics [2, 9, 10]. The impact of changes in ecosystem functionality resulting from exploitation

**Data Availability Statement:** The datasets utilized in this study are accessible through various online repositories. Species traits data are available for download from the Zenodo repository at https://

doi.org/10.5281/zenodo.10255369. Biomass data can be obtained from the Azores Fisheries Auction Services at https://www.lotacor.pt/pescado-descarregado. The code for performing the null models and fuzzy correspondence analysis (FCA) is available on GitHub at https://github.com/Eudriano/Trait_diversity.git.

**Funding:** This work was performed under the framework of the project FunAzores co-funded by AÇORES 2020, through the FEDER fund from the European Union: ACORES 01-0145-FEDER-000123. Okeanos team received national funds through the FCT – Foundation for Science and Technology, I.P., under the project UIDB/05634/2020 and UIDP/05634/2020 and through the Regional Government of the Azores through the initiative to support the Research Centers of the University of the Azores and through the project M1.1.A/REEQ.CIENTÍFICO UI&D/2021/010. AC is supported by the national funds through the FCT within the scope of CEECIND/ 00101/2021 and https://doi.org/10.54499/2021.00101.CEECIND/CP1669/CT0001 The funders had no role in study design, data collection and analysis, decision to publish, or preparation of the manuscript.

**Competing interests:** The authors declare that the research was conducted in the absence of any commercial or financial relationships that could be construed as a potential conflict of interest. This does not alter our adherence to PLOS ONE policies on sharing data and materials.

extends beyond individual species or habitats and can have profound implications for marine biodiversity, ecological processes, and ecosystem services [3, 11, 12]. Despite the potential loss of ecosystem functions, the predominant approach to fisheries management still revolves around single-species stock assessments, primarily targeting economically valuable species, while overlooking the interconnectedness of species and the ecosystem-wide impacts of fishing activities [3, 12–15].

Trait-based approaches has gained recognition in fisheries sciences, offering valuable insights into fisheries activities and ecosystem processes [7, 15–20]. This approach relies on understanding the traits that species possess such as body shape, trophic position, diet, size at first maturity, fecundity, and how these traits contribute to ecosystem processes, influencing their interactions with the environment and other organisms [11, 21, 22]. The trait diversity reflects the range of functional roles and ecological niches occupied by different species within an ecosystem [11, 21]. For instance, traits associated with species' food acquisition, reproduction and habitat preferences can determine the roles species play in nutrient cycling, energy transfer, and trophic interactions within ecosystems [11]. Understanding ecosystem functioning using trait diversity as proxy can provide valuable insights into the potential impacts of exploitation on specific ecological functions played by targeted species by fisheries [11, 12]. As fisheries target specific species based on their economic value or abundance, changes in the trait diversity of caught assemblages can provide insights into potential shifts in ecosystem structure and functioning [12, 17, 22]. For instance, in a novel study conducted by Mbaru et al. [15], it was found that all fishing gears targeted a wide diversity of traits, but with some differentiation among gears. These authors suggested that monitoring specific gears (e.g., hook and lines) could be crucial for preserving targeted species with rare trait combinations that play particular ecological roles within the ecosystem. Thus, the impacts of fishing activities on ecosystem function can be assessed by observing changes in functional diversity metrics, and trait diversity of exploited fish assemblages [7, 12, 15, 23].

Fisheries are crucial in the Azores Archipelago, which, despite its rich marine biodiversity, has limited fishing grounds with less than 1% of its EEZ being shallower than 600 m and an average depth of 3000 m [24, 25]. Since the 1980s, the Azorean fishing industry has expanded from island shelves to offshore seamount areas and deeper waters, driven by advancements in air transport of fresh fish and high-tech equipment on boats, such as sonar [26–28]. Classified as small-scale, about 60% of Azorean vessels are under nine meters in length, and small-scale fishing comprises 80–90% of the fleet [26–29]. This fleet targets various species, including tuna, deep-water demersal, and small pelagic species, across different depths [20, 24, 28, 30, 31]. Fish are commercialized through local and coastal fleets: the local fleet operates small boats (under 12 meters) near coastal areas up to 700 m deep, while the coastal fleet targets deeper waters over 700 m [27, 32]. Both fleets primarily use selective hook and line gears, such as handlines and longlines, with traps and small purse-seine methods also used, especially by local operations [27, 28].

The small-scale fisheries in the Azores are recognized for their sustainable practices, supported by an efficient fishery data collection system established since the 1970s and local regulations [33, 34]. These regulations include fishing closures (e.g., ordinance n° 74/2015), species-specific quotas, and minimum landing sizes (e.g., ordinance n° 21/2019). Limits are set for annual landings of species like *Phycis phycis*, *Helicolenus dactylopterus*, *Serranus atricauda*, and *Pagellus bogaraveo*, with a ban on targeting sharks *Galeorhinus galeus* and *Prionace glauca* (e.g., ordinances n° 92/2019 and n° 27/2023). Despite efforts for sustainable resource use, knowledge gaps remain regarding the potential impacts of fishing on marine biodiversity and ecosystem functions. In an effort to address this concern, the present study aimed to investigate the trait diversity of fish landed by both local and coastal fishing fleets in the Azores

archipelago over the past four decades using null models. This trait-based approach allowed us to test two main hypotheses: first, that local and coastal fishing fleets have impacted the diversity of fish traits landed in Azorean islands differently over the past four decades, and second, that the adoption of highly selective gears by these fishing fleets has reduced trait diversity during this time frame. Additionally, discussing the implications of the findings for fishery management enables informed decision-making that considers the broader ecological consequences of fishing activities.

## 2. Material and methods

### 2.1 Study area

Nestled within the Macaronesia region of the North Atlantic Ocean, the Azores archipelago is a group of nine volcanic islands, integrated into the broader open ocean ecosystem along the Mid-Atlantic Ridge. The archipelago's Economic Exclusive Zone (EEZ) encompasses a rich expanse of marine life, attracting fishing operations that focus on both nearshore areas and the 461 identified seamounts scattered throughout the region [24]. These seamounts serve as hotspots for diverse fish species, creating a vibrant tapestry of aquatic biodiversity [24–26].

### 2.2 Ladings data

Landing data of the fish species exploited in the Azores archipelago from 1980 to 2020 were provided by the Azores Fisheries Auction Services (LOTAÇOR/OKEANOS-UAc) (www.lotacor.pt/). This landing dataset included the total of fish landed in weight, common name of the species, month, year and details about the type of fishing fleet. It distinguished between local and coastal fishing vessels responsible for landing each species in their respective year. The scientific names of the reported species in the annual landings were cross-referenced with the studies published by Santos et al. [27] and Santos et al. [28]. Experts specializing in Azorean fish species also assisted in verifying the accuracy of the species names. All fish species included in this study are listed in **S1 Table**. For more information about all fish species reported in the Azorean landings, as well as the total landings by species, can be found in Costa et al. [20], and Costa et al. [29].

### 2.3 Functions and trait modalities

Fourteen functional traits were chosen based on the biological and ecological knowledge of species that are critical for ecological processes in marine ecosystems [20]. Trait data were obtained from Costa et al. [20] and Costa et al. [30], primarily sourced from FishBase [31], and supplemented with information from scientific literature. Swimming mode and diet traits were added to the original trait database. The modalities of the former trait were assigned based on FishBase [31], and Trindade-Santos et al. [17], whereas the later trait according to FishBase and scientific literature [32]. The prey items, representing dietary components, were used to assign diet modalities. Species were assigned to multiple modalities, if necessary, as this method more accurately depicts the diversity and overlap of species diets [22].

Ordinal and continuous trait variables were converted into categorical traits. This conversion aimed to mitigate the impact of unknown trait values, which were approximated using trait values from closely related species. Additionally, it aimed to standardize the estimation methods for fecundity (e.g., vitellogenin oocytes, hydrated oocytes, eggs, and embryos), size (e.g., total length, standard length, and fork length), and size at first maturity [22, 30]. The **Table 1** provides a comprehensive overview of all traits, their respective modalities, and the potential functions of the species within the ecosystem.

**Table 1. Fish species traits, functions and modalities use to assign fish species landed in the Azores archipelago.**

| Trait | Function | Modalities | |
|---|---|---|---|
| Position in water column | Habitat use | Bathydemersal (Batd) | Pelagic neritic (Pene) |
| | | Bathypelagic (Batp) | Pelagic oceanic (Peoc) |
| | | Benthopelagic (Benp) | Reef associated (Reas) |
| | | Demersal (Deme) | |
| Mean temperature preference | Habitat use | <10˚C | 20–25˚C |
| | | 10–15˚C | >25˚C |
| | | 15–20˚C | |
| Maximum depth | Habitat use | <500 m | 1500–2000 m |
| | | 500–1000 m | >2000 m |
| | | 1000–1500 m | |
| Swimming mode | Locomotion | Ammiform (Amii) | Rajiform (Raji) |
| | | Anguilliform (Angu) | Subcarangiform (Subc) |
| | | Balistiform (Bali) | Tetraodontiform (Tetr) |
| | | Carangiform (Cara) | Thunniform (Thun) |
| | | Labriform (Labr) | |
| Body shape | Habitat use Locomotion | Eel-like (Eeli) | Fusiform (Fusi) |
| | | Elongated (Elon) | Short/ deep (Shde) |
| | | Flattened (Flat) | |
| Maximum body size | Feeding | <100 cm | 300–400 cm |
| | Life history | 100–200 cm | >400 cm |
| | Locomotion | 200–300 cm | |
| Trophic position | Habitat use | <2.5[†] | 3.5–4.5 |
| | Feeding | 2.5–3.5 | >4.5 |
| Diet | Habitat use | Benthic invertebrates (Binv) | Detritus (Detr) |
| | Feeding | Cephalopods (Ceph) | Plant and algae (Herb) |
| | | Chondrichthyans (Chon) | Teleosts (Tele) |
| | | Crustacean decapods (Crde) | Zooplankton (Zoop) |
| Generation time | Life history | <5 years | 10–15 years |
| | | 5–10 years | >15 years |
| Growth coefficient | Life history | >0.3[‡] | 0.6–0.9 |
| | | 0.3–0.6 | >0.9 |
| Food consumption | Habitat use | <5 Q/B[*] | 10–15 Q/B |
| | Feeding | 5–10 Q/B | >15 Q/B |
| Size at first maturity | Life history | <50 cm | 150–200 cm |
| | | 50–100 cm | >200 cm |
| | | 100–150 cm | |
| Reproductive guilds | Life history | Bearers (Bear) | Mixed (Mixe) |
| | | Guarders (Guar) | Nonguarders (Ngua) |
| Fecundity | Life history | <10[•] | $10^4–10^5$ |
| | | 10–10 | $10^5–10^6$ |
| | | $10^2–10^3$ | >$10^6$ |
| | | $10^3–10^4$ | |

[†] It represents the position that fish occupy within their respective food webs [31].

[‡] This is a parameter of the von Bertalanffy growth function (also known as the growth coefficient), expressing the rate (1/year) at which the asymptotic length is approached. It quantifies how quickly an organism grows towards its maximum possible size over time [31].

[*] Q/B is the food consumption per unit biomass [31].

[•] It refers to the number of oocytes in the most developed stages, eggs or embryos.

The functional traits corresponded to four fundamental fish functions: habitat use, locomotion, feeding and life history, encompassing a total of 76 trait modalities. Each function was represented by a minimum of four trait modalities (**Table 1**). Thus, habitat use plays a crucial role in determining the distribution, abundance, and diversity of fish species, shaping their interactions within the ecosystem [33–36]. Locomotion is essential for the survival and fitness of fishes, playing a key role in food acquisition and enhancing their ability to evade predators [37–39]. Feeding is a critical aspect for fish, as it serves as the primary means for obtaining energy needed for growth, development, and reproduction [40]. Feeding behavior holds significant ecological importance, as it shapes trophic interactions, nutrient cycling, and both inter-specific and intraspecific competition within ecosystems [41–43]. Life-history traits, such as growth rate, size at maturity, generation time, and fecundity, play a crucial role in influencing the reproductive success and survival of individual fish, as well as shaping the dynamics of fish populations [44–46].

## 2.4 Functional diversity metrics

It's crucial to clarify the concepts of functional entities (FEs) and functional trait space for understanding their roles in the context of metrics assessing functional diversity. The FEs refer to unique combination of trait categories within a landed fishing assemblage. Species with similar trait values are clustered together, and these shared FEs often indicate similar ecological niches and functional roles within the ecosystem [47–49]. On the other hand, the trait space, also known as functional space, is a multidimensional conceptual space where the axes represent functional traits, and species or FEs are positioned within this space based on their respective trait values (**Fig 1**) [50]. Species possessing uncommon combinations of trait are positioned toward the border of the trait space and are considered more functionally specialized, playing different ecological roles in the ecosystem compared to those positioned in the center of the trait space [20, 51].

Four functional diversity metrics, including functional richness (FRic), functional evenness (FEve), functional divergence (FDiv), and functional dispersion (FDis), were used to assess changes in traits diversity of fish species reported in the landings over time. The FRic metric measures the amount of functional trait space occupied by the fish in the landed assemblage over a specific period [51]. High FRic indicates that fishing fleet landed species with a greater variety of functional traits (**Fig 1A**). FEve indicates how evenly the combination of traits of different species is distributed in terms of landings. Low FEve indicates that fisheries target a group of fish with a particular combination of traits (**Fig 1B**) [17, 20]. FDiv measures the distance of highly landing species with extreme trait combinations from the gravitational center of the trait space [51]. A high FDiv suggests that the highest landed species, in terms of catch, possess uncommon trait combinations or represent more functionally specialized species (**Fig 1C**). In other words, fisheries remove a high biomass of species that play particular roles in the ecosystem. FDis measures the mean distance of individual species to the centroid of all species in the trait space [52]. High FDis suggests that the highest landed species belong to a specific group positioned toward the periphery of the trait space relative to the centroid of traits (**Fig 1D**). See Mason et al. [53], Villéger et al. [51], and Laliberté and Legendre [52] for more detailed information about the functional metrics.

## 2.5 Data analysis

The data analysis was conducted using R statistical software (version 4.0.0; R Foundation for Statistical Computing) (R Core Team, 2020). Data manipulation, including processing,

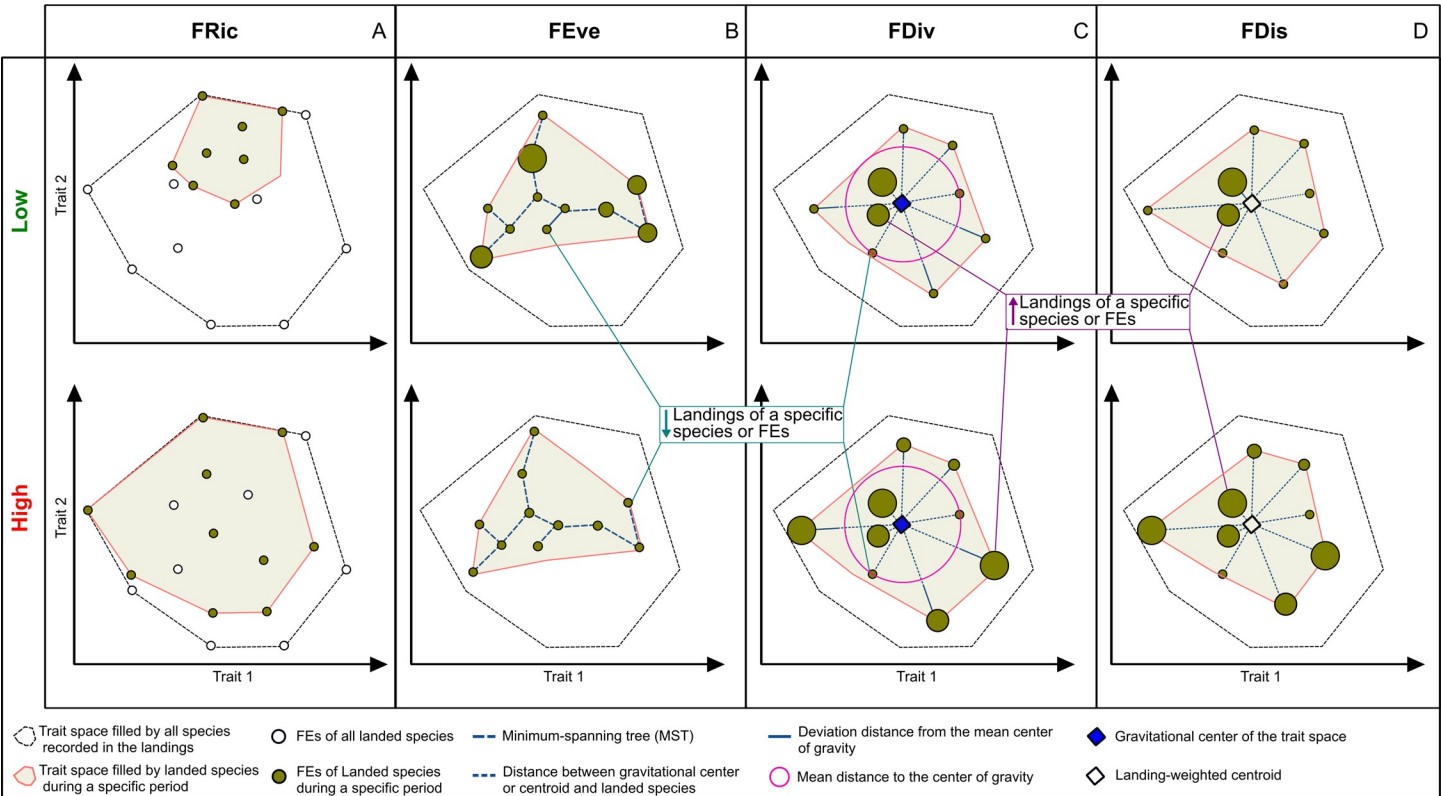

**Fig 1. Conceptual diagrams illustrating the functional diversity metrics (based on Villéger et al. [51], and Laliberté and Legendre [52].**

organizing, and cleansing, was performed using functions from the 'tidyrverse' [54], 'tibble' [55], 'stringr' [56], and 'dplyr()' [57] packages.

**2.5.1 Coding of traits.** The trait data consisted of two matrices. The first matrix involved the transformation of the categorical traits into binary format. The second matrix was created using the diet binary data, with trait modalities (food items) as columns and each species as rows, considering that each species was assigned to multiple food items. Hence, species' affinities to trait modalities were represented by frequencies, ensuring that the sum of each row, per trait per species, equaled one (**see example in S1 Fig**) [58, 59]. This transformation was performed using the 'prep.fuzzy()' function from the R package ade4 [60, 61]. Both binary and fuzzy-coded matrices were merged to create the final matrix, called bin-fuzzy matrix, which was used for both the computation of functional diversity metrics and fuzzy correspondence analyses (FCA).

**2.5.2 Functional diversity metrics (FD).** The FD metrics were calculated using the 'dbFD ()' function from the R package FD [62]. This function requires two matrices as input: a distance matrix and a species occurrence matrix. The former matrix was created from the bin-fuzzy matrix using the function 'dist.ktab()' from the R package ade4 [60, 61]. The latter matrix contained the species landings and was standardized to a range 0–1 scale. This standardization was performed using the formula $x' = x- \min(x)/[\max(x)-\min(x)]$, where $x$ represents the original value and $x'$ represents the standardized value [50, 63].

The dbFD function uses Principal Coordinates Analysis (PCoA) to compute the FRic, FEve, FDiv and FDis metrics [51]. The number of PCoA axes retained for calculating the FD metrics was determined as "min" value, defined as the maximum number of traits that satisfies

the condition $s> = 2\text{^}t$, where s represents the number of species and $t$ represents the condition to be met. FRic was standardized by the global FRic, which included all species, constraining FRic between 0 and 1. For FEve, FDiv and FDiv, the dbFD function was set to be weighted by abundance, i.e., landings, of the species.

**2.5.3 Null models.**   Null models were performed to assess the significance of each diversity metrics (FRic, FEve, FDiv and FDis) between fishing types (local and coastal) and across decade pairs (80s-90s, 80s-00s, 80s-10s, 90s-00s, 90s-10s, and 00s-10s). These assessments considered habitat use, locomotion, feeding and life history functions. In this process, the binary trait matrix was used to simulate the randomization of trait matrices and assess their impact on each functional metric. The randomization procedure shuffled species names to ensure that each species retains the original trait values. This approach allows exploration of different species assemblages while considering that certain trait combinations may not naturally coexist. Randomization was repeated 999 times to obtain a simulated distribution for each metric under the null hypothesis of random species arrangement. The observed value was then compared to this distribution to assess its statistical significance. Hence, absolute differences between observed and simulated values were calculated. The counts of simulated differences exceeding the observed differences were then determined, and the p-values calculated by dividing the exceedances by the total number of iterations in the randomization process (999). The R code for this entire process can be found in Costa [64].

**2.5.4 Fuzzy correspondence analysis (FCA).**   The FCA was performed to analysis the association between the trait modalities and the species by function in the trait space. This analysis utilized the 'dudi.fca()' function from the ade4 package [58, 60, 61]. The first four coordinates of the FCA axes were extracted and used to plot the functional trait space for habitat use, locomotion, feeding and life history functions. In addition, the 'inertia.dudi()' function also from ade4 package was employed to compute the decomposition of inertia, measuring the contribution of the modalities to each axis. A Person correlation test was applied to evaluate the relationship between each trait modality and FCA axes, using the FCA species coordinates. The R code for this entire process, including the FCA and trait space plots, can be found in Costa [64].

## 3. Results

### 3.1 Functional diversity metrics and modalities

The Analysis of combined functions revealed no significant differences in FRic, FEve, FDiv and FDiv between the two fishing types (**S2 Table**). This pattern was also observed across all four functions, corroborating the idea of similar trait diversity between assemblages landed by local and coastal fishing (**Table 2**). FRic did not vary significantly across functions and decades for both fishing fleets (**Figs 2A and 3A**). However, metrics commonly influenced by landings, such as FEve, FDiv and FDis, exhibited significant differences between certain decade pairs concerning the fishing fleet and functions such as habitat use, locomotion and feeding. None of these three metrics showed significant changes for the life history function over decades.

In the case of local fishing, FEve was significant only for feeding, observed between 2000s and 2010s (**Fig 2B**), whereas FDiv was significant for locomotion and feeding functions, observed between 1980s and 1990s, and 1990s and 2010s, respectively (**Fig 2C**). FDis exhibited significance for locomotion and feeding in two different decades pairs (**Fig 2D**). In species landed by coastal fishing, FEve was significant only for feeding function observed in the decade pairs 80s-00s, 90s-00s, and 00s-10s (**Fig 3B**). On the other hand, FDiv and FDis did not show significance across functions and decades (**Fig 3C–3D**). Nevertheless, marginal significance (p = 0.0641) was observed for FDiv between the 1990s and 2010s for locomotion function

**Table 2. Functional diversity metrics for fish landed in the Azores archipelago over the past four decades.** nbsp: number of species, sing.sp: species functionally different in the landings, quali.FRic: quality of the reduced-space representation required to compute FRic and FDiv, FRic: functional richness, FEve: functional evenness, FDiv: functional divergence, FDis: functional dispersion. P-values represent significance between local and coastal pair for each metric in each analysis from randomization testing, considering each function (habitat use, locomotion, feeding and life history).

| | Local | Coastal | Habitat use | | Local | Coastal | Locomotion | |
| --- | --- | --- | --- | --- | --- | --- | --- | --- |
| | | | Differences | p-value | | | Differences | p-value |
| nbsp | 103 | 100 | | | 103 | 100 | | |
| sing.sp | 96 | 94 | | | 32 | 32 | | |
| quali.FRic | 0.593 | | | | 0.7641 | | | |
| FRic | | | 0.0185 | 0.2452 | | | 0.0000 | 0.4384 |
| FEve | | | 0.0591 | 0.1331 | | | 0.0181 | 0.5846 |
| FDiv | | | 0.0694 | 0.2723 | | | 0.0301 | 0.6527 |
| FDis | | | 0.0085 | 0.8529 | | | 0.0121 | 0.8028 |
| | Local | Coastal | Feeding | | Local | Coastal | Life history | |
| | | | Differences | p-value | | | Differences | p-value |
| nbsp | 103 | 100 | | | 103 | 100 | | |
| sing.sp | 62 | 62 | | | 61 | 62 | | |
| quali.FRic | 0.6945 | | | | 0.6606 | | | |
| FRic | | | 0.0004 | 0.7538 | | | 0.0000 | 0.8669 |
| FEve | | | 0.0363 | 0.3263 | | | 0.0035 | 0.9289 |
| FDiv | | | 0.0715 | 0.3263 | | | 0.0152 | 0.8248 |
| FDis | | | 0.0084 | 0.9039 | | | 0.0167 | 0.7437 |

(**Fig 3C**). Detailed results of all simulation analysis for both local and coastal fishing, comparing each metric for decade pairs across functions can be found in **Costa et al.** [**32**].

## 3.2 Fuzzy correspondence analysis

The correlation between the FCA trait vectors and its first two axes varied based on habitat use, locomotion, feeding and life history functions, as well as the distribution of the landed species in the functional space. Overall, the inertia explained by the combination of the two first FCA axes (axis 1 and axis 2) ranged from 22.5% to 30.5% (**S5–S8 Tables**).

In the FCA related to habitat use, the first two FCA axes explained 22.5% of the inertia. The highest positive correlations ($r^2 > 0.50$) between the traits and axis 1 were observed for maximum depth less than 500 m, and fish with a fusiform body shape. Conversely, the strongest negative correlations ($r^2 > -0.62$) were observed for temperature preference less than 10°C, food consumption less than 5, and bathydemersal species (**Fig 4A**). However, the fifteen most landed species occupied different positions in the functional space, with variations in their habitat utilization (**Fig 4B**). *Sparisoma cretense* was notably positioned uniquely at the vertices of the functional space, showing significant differences in habitat utilization compared to other highly landed species (**Fig 4B**). The correlation test results performed to evaluate the relationship between all traits and the four FCA axes for FCA habitat use can be found in **S5 Table**.

In the locomotion FCA function plot, species with eel-like shaped bodies and those swimming in the anguilliform mode exhibited the strongest positive correlations with the first FCA axis ($r^2 > 0.50$). Species with flattened bodies swimming in the rajiform mode showed the strongest positive correlation with FCA axes 2 ($r^2 > 0.90$). The highest negative correlation (-0.54) was observed for fusiform species with FCA axis 1 (**Fig 4C**). In general, most species were clustered closer to the center of the biplot, with an unclear association between the species and specific functional traits related to locomotion type. However, there were exceptions for flattened fish and eel-like species such as *Conger conger* and *Lepidopus caudatus* (**Fig 4D**).

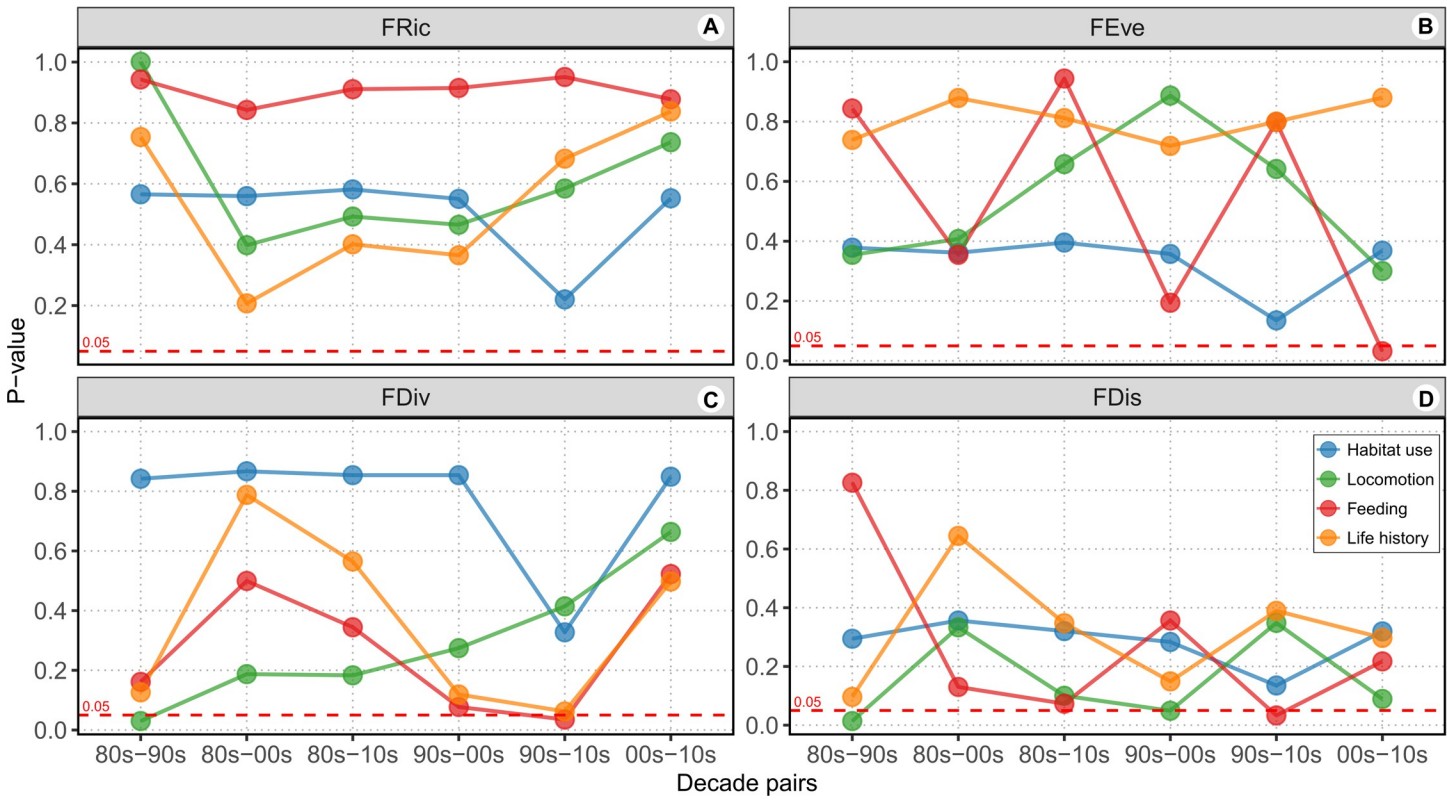

**Fig 2. Functional diversity metrics for fish landed by local fishing in the Azores archipelago for four decades.** The y-axis p-values indicate the significance among decade pairs for each functional metric (FRic: functional richness, FEve: functional evenness, FDiv: functional divergence, and FDis: functional dispersion) in each analysis conducted through randomization testing. The analysis considered the functional modalities habitat use, locomotion, feeding, and life history. The dashed red line represents the p-values at 0.05, where values below this line indicate significance of the metric between decade pairs in relation to the trait function. For details see S3 Table and Costa et al. [32].

*L. caudatus* exhibited the most unusual combination of locomotion traits among Actinopterygii, positioning as a vertex species (Fig 4D). The correlation test results performed to evaluate the relationship between all traits and the four FCA axes for FCA locomotion can be found in S6 Table.

The feeding FCA exhibited relationships among trait vectors, dividing the trait space in two groups, primarily driven by differences in body size, trophic position and diet. The feeding function plot showed a positive correlation between the FCA axis 1 and the species with body size less than 100 cm ($r^2 = 0.72$), trophic position between 2.5 and 3.5 ($r^2 = 0.66$), and herbivorous species ($r^2 = 0.62$). The strongest negative correlations ($r^2 > 50$) were observed for food consumption less than 5, trophic position between 3.5 and 4.5, and species that prey on cephalopods and teleosts. Species with trophic position less than 2.5, and food consumption between 10 and 15 were positively correlated with the FCA axis 2 ($r^2 > 0.55$) (Fig 4E). The scattered arrangement of the most landed species in the functional space reveals a variety of functional traits associated with feeding function, encompassing variations in maximum size, trophic position, food consumption and diets (Fig 4F). Additionally, the vertex species *S. cretense* displayed an unusual combination of feeding-related traits (Fig 4F). The correlation test results performed to evaluate the relationship between all traits and the four FCA axes for FCA feeding can be found in S7 Table.

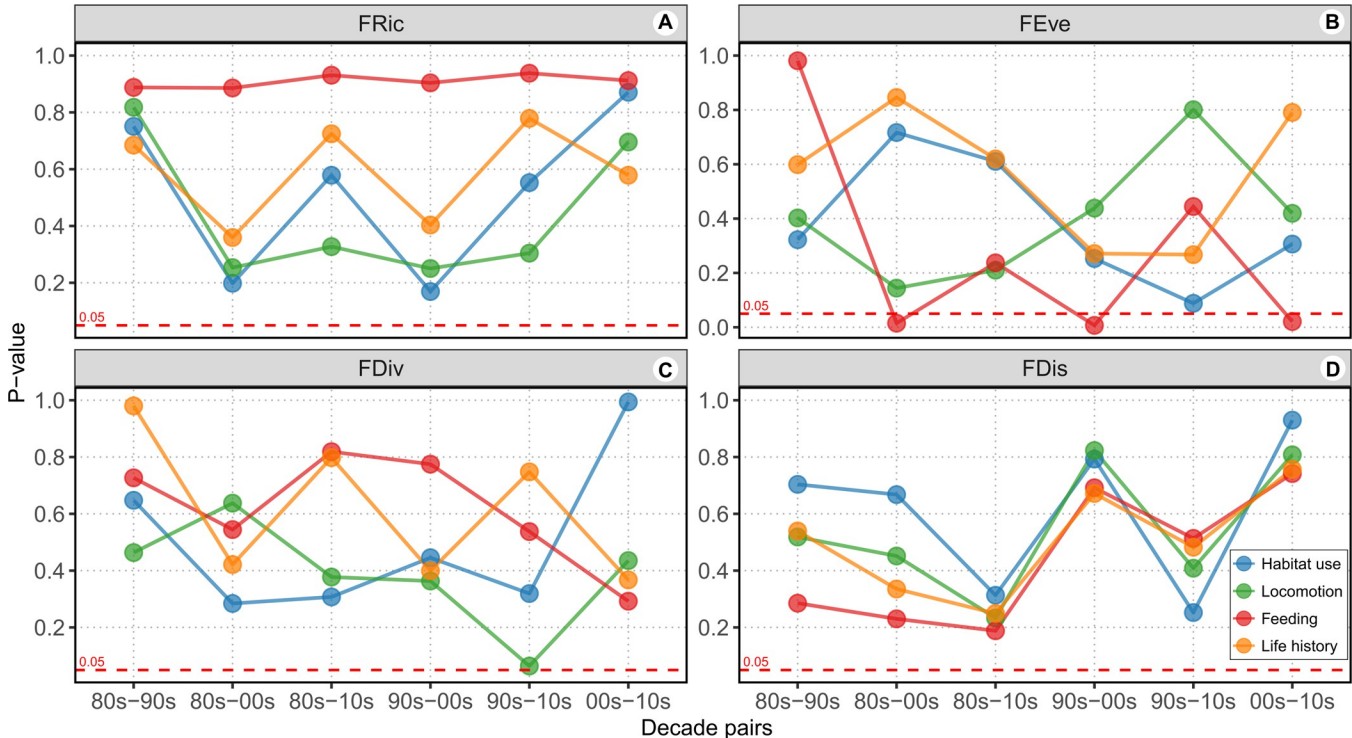

**Fig 3. The y-axis p-values indicate the significance among decade pairs for each functional metric (FRic: Functional richness, FEve: Functional evenness, FDiv: Functional divergence, and FDis: Functional dispersion) in each analysis conducted through randomization testing.** The analysis considered the functional modalities habitat use, locomotion, feeding, and life history. The dashed red line represents the p-values at 0.05, where values below this line indicate significance of the metric between decade pairs in relation to the trait function. For details see **S4 Table** and Costa et al. [32].

In the function plot for life history, the highest positive correlation values between traits and axis 1 were observed for bearers species with high generation time (>15 years), large body size (>400 cm), high size at first maturity (>200) and low fecundity (< 10). Conversely, the strongest negative correlations ($r^2$< -0.50) observed for this same axis were associated with fish that reach the first sexual maturity at less than 50 cm, exhibit a maximum body size less than 100 cm, have a generation time less than 5 years, and do not guard their eggs (nonguarders). Regarding to the axis 2, positive correlation higher than 0.50 was recorded for generation time between 5 and 10 years, a body size between 100 and 200 cm, and growth coefficient less than 0.54/ year (**Fig 4G**). In general, the negative correlations were associated with the majority of species landed, clustered towards to the left border of the functional space (**Fig 4H**). The Elasmobranchii *Dalatias licha* exhibited an unusual combination of traits associated with life history as a vertex species, contrasting with the traits observed in the most landed species (**Fig 4H**). The correlation test results performed to evaluate the relationship between all traits and the four FCA axes for FCA life history can be found in **S8 Table**.

## 4. Discussion

The present study represents the first attempt to connect trait diversity to fish landings over time through a comprehensive analysis using null models. By integrating trait diversity metrics with fisheries data, this study provides novel insights into the influence of fishing activities on the functional composition of fish assemblages and ecosystem dynamics. Understanding the differences in trait diversity and identifying potential changes in ecological functions over time within the

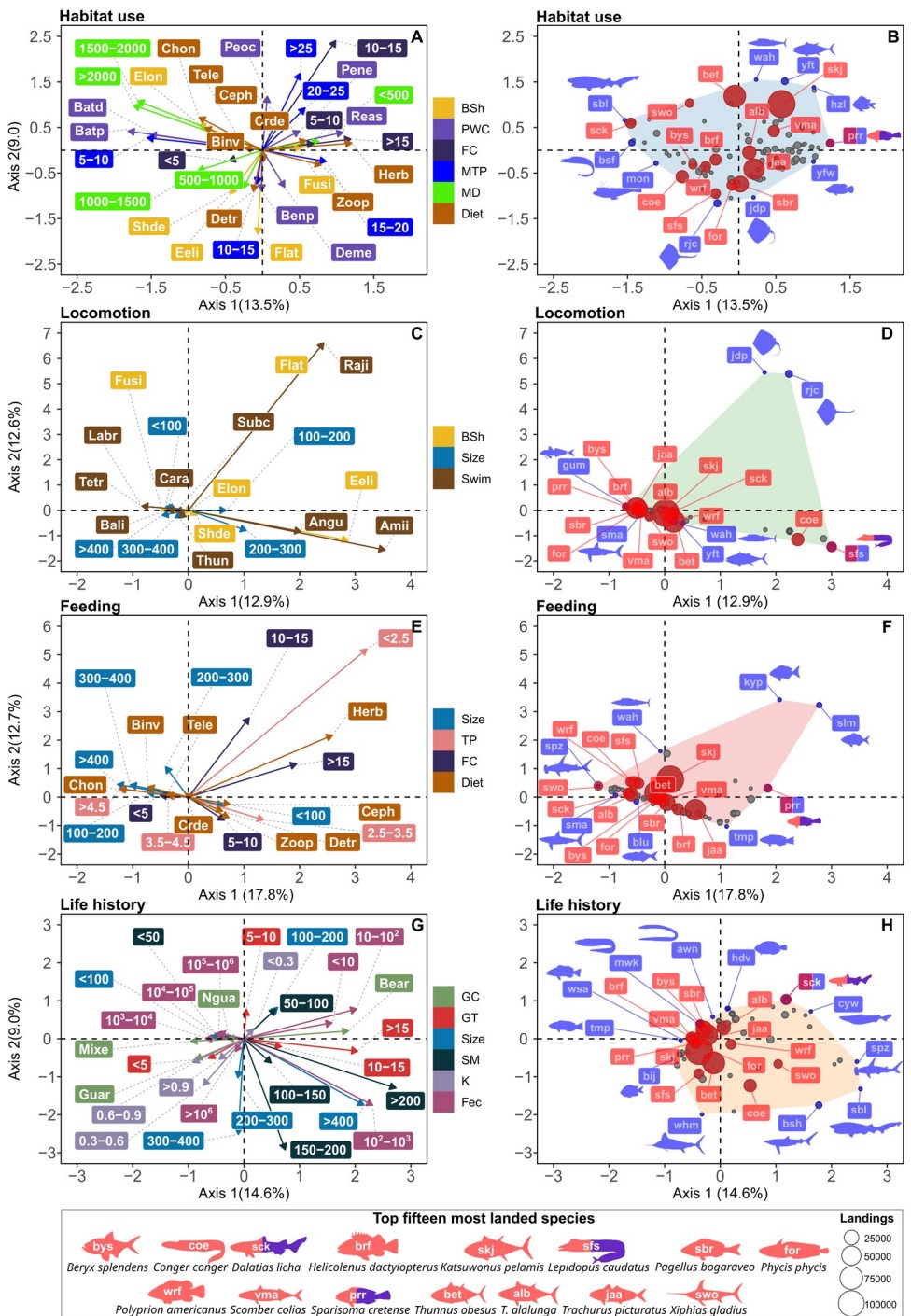

**Fig 4. Exploring the relationships between functional traits and fish species within each function (habitat use, locomotion, feeding, and life history) in the functional space (areas in blue, green, light red, and light orange).** The plots of the fuzzy correspondence analysis (FCA), .i.e., A, C, E and G, illustrate the correlation between the FCA axes (axes 1 and 2), and functional trait modalities (See also **S5–S8 Tables and S1 Fig**). Different colors represent various trait modalities in FCA plots. In the plots B, D, F, and H, the fifteen most landed species and their total landings in tonnes are highlighted in red. Abbreviation codes available in **Table 1**. The scientific names of each vertex species highlighted in blue can be seen in **S9 Table**.

fisheries context are crucial for providing fisheries managers with insights into ecosystem structure and potential responses to exploitation resulting from trait diversity [15, 17, 20, 53].

## 4.1 Diversity metrics, functions, and trait diversity

Fishing activities can influence diversity metrics and alter functional traits within ecosystems by changing species composition and biomass landed [7, 15, 17, 20, 71]. Costa et al. [20] reported significant variations in FRic and functional space of fish species landed in the Azores over the past three decades, attributed to variations in species with unusual trait combinations in annual landings. Costa et al. [36] also found significant changes in species composition landed by both local and coastal fishing in the Azores, affecting diversity metrics such as functional redundancy (FRed) and functional vulnerability (FVul). However, these studies did not comprehensively investigate trait diversity. The present study fills this gap using null models, revealing no significant differences in FRic between fishing types and over time across functions, suggesting that Azorean fisheries have targeted species with a consistent range of traits through both local and coastal fishing over the past four decades.

Despite the absence of significant variations in trait diversity across decades, the species landed by Azorean fishing fleets exhibit a wide range of trait combinations, indicating occupation of diverse ecological niches. This diversity is evident even among the fifteen most landed species. In Azorean waters, fish are primarily caught using various handlines and longlines, while small pelagic species such as *Trachurus picturatus*, *Scomber japonicus*, and *Boops boops* are caught using purse-seine nets [27, 28, 72]. Mbaru et al. [15] suggested that selective gear types target a narrower range of functional diversity, potentially impacting fewer ecosystem functions. In the Azores, most selective gear types are associated with bottom habitats, capturing species such as *Raja clavata*, *P. bogaraveo*, *C. conger*, *H. dactylopterus*, *Mora moro*, *P. phycis*, and *Beryx* spp. [72, 73]. Menezes et al. [24] reported that fish assemblages in the Azores are closely related to depth, with three main depth-related discontinuities (<200 m, 200–600 m, and >700 m). The type of fishing fleet is also related to specific fishing areas and depths, with coastal fishermen using longlines typically catching fish at deeper waters (200–700 meters) compared to the local fishing fleet [24, 32]. The disparity between our findings and those of Mbaru et al. [15] can be attributed to the species diversity and variability in bathymetric composition of fish species landed in the Azores, contributing to the diversity of trait combinations among landed species.

The amount of landed fish can influence the types of trait combinations removed from the ecosystem by fisheries over time, affecting metrics such as FEve, FDiv, and FDis [7, 18–20]. Although trait diversity did not change over the past 40 years in Azorean waters, significant differences were found between some decade pairs in FEve, FDiv, and FDis in both coastal and local landings. Exploiting a broad array of species with diverse ecological roles can change these metrics [17]. Costa et al. [20] reported increased landings of species such as *Trachurus picturatus*, *Katsuwonus pelamis*, and *Thunnus obesus*, contributing to a decrease in FEve over the last two decades in the Azores. Variability in FDiv was also linked to *K. pelamis*, which had unusual trait combinations compared to other landed species in recent decades. Costa et al. [36] noted differences in the ten most landed species between fishing types in the Azores, with *T. picturatus*, *P. bogaraveo*, and *K. pelamis* being most landed by local fishing, while *K. pelamis*, *T. obesus*, and *T. picturatus* dominated coastal fishing. Landings often dominated by a few taxa can impact diversity metrics weighted by catch data [18, 19, 71, 73, 74]. Thus, the significant differences observed in FEve and FDis between local and coastal fishing, particularly in Figs 3C and 4B, may result from temporal variability in the catches of the most landed species with unusual trait combinations compared to the other landed species.

The prevalence of species belonging to a single functional entity indicates both a high level of specialization and a significant trait diversity [47, 65]. This finding aligns with the conclusions of Costa et al. [30], who reported low FRed and high FVul for both all fish species recorded in Azores and those targeted by Azorean fishing fleets. These authors revealed that over 70% of species in the landings occupied a single functional entity, suggesting that the Azorean fish fauna is highly susceptible to a decline in ecosystem functions, particularly due to fishing activities. Similarly, Rincón-Diáz et al. [7], found a high number of functional entities with only one species among fish species exploited in the Argentinian continental shelf, indicating low redundancy, with each ecological role typically fulfilled by a single fish species. This finding highlights the importance of minimizing the impacts of fishing activities on ecological functions and processes performed by single-FE species. For instance, the overexploitation of top predators (trophic level above 4) such as species *D. licha* (Elasmobranchii), *Beryx* spp., *C. conger*, and *P. bogaraveo* can lead to an increase in abundance of their prey populations, potentially disrupting predator-prey interactions and reshaping community structures [9, 10, 66]. Thus, the presence of the single-FEs species ensures the resilience of the ecosystem by preserving key ecological functions [15, 22]. Implementing fishery measures such as quotas, minimum landing size, area, and temporal fishing closures (e.g., local regulation ordinance nº 74/ 2015) is crucial to mitigate the impact of fishing on ecosystem functions and processes in Azores.

## 4.2 Fuzzy correspondence analysis

The species landed by both local and coastal fishing fleets exhibited high dispersion in the functional space regarding habitat use function, indicating a wide range of trait combinations within the landed catch. Nevertheless, many species were clustered, primarily driven by fusiform-shaped species inhabiting shallower waters. This finding suggests two interesting patterns in habitat utilization. Firstly, the landed species comprise species with diverse habitat preferences, ranging from shallower coastal waters to deeper environments. Notably, among the fifteen most important deep-sea species (with a maximum depth distribution deeper than 600 m) reported in the landings were *Beryx* spp., *Polyprion americanus*, and *H. dactylopterus*. Secondly, the clustering of species may indicate that certain traits are more prevalent or dominant within the landings, such as a fusiform body shape and environments with a maximum depth less than 500 meters. Among the fifteen most landed species from shallower waters exhibiting a fusiform body shape were *T. picturatus*, *S. cretense*, and *K. pelamis*. This diversity in habitat utilization can be attributed to the local and coastal fishermen, who visit various habitats in terms of depth [24, 67, 68]. This finding also aligns with the depth-related ichthyofaunistic discontinuities reported in the Azores, as discussed previously [25, 69].

Fishing pressure can exert selective pressure on the body shapes and sizes of target species, and these changes can subsequently affect locomotion-related traits and functions [30, 70, 71]. These traits play a critical role in controlling the food web structure, impacting diversity and interactions with other organisms within the water column [72–74]. Alós et al. [70] reported that individuals in the population of both species with larger mouths and more streamlined and elongated bodies were found to be more vulnerable to hook-and-line, in turn creating selection for smaller mouth and deeper bodies. The results of the present study partially corroborate this idea, as Azorean fisheries primary adopt high selective fishing methods (hook and lines), and most individuals caught were associated to fusiform body shape. However, further research is needed to refine and validate this approach. In addition, differences in locomotion trait space were primarily driven by few species with eel-like and flattened body shapes. The observed differences in locomotion traits highlight the ecological significance of eel-like

and flattened body shapes within the fish assemblages landed in the Azores. This suggests that eel-like species such as *C. conger*, and *L. caudatus*, as well as flattened species like *Dasyatis pastinaca* and *R. clavata*, play important roles as specialists in the ecosystem dynamics, based on their specialized locomotion strategies and associated habitat preferences.

Species with specialized feeding approaches may play key roles in shaping food webs, energy flow, and nutrient cycling processes, thereby influencing ecosystem structure and function [9, 10, 22]. In the Azorean waters, the feeding function results revealed that variability in species trait combinations was primarily driven by differences in body size, trophic position and diet. This suggests the fish grouped on each side of the FCA play similar functional roles in terms of feeding. Specifically, species such as *Xiphias gladius*, *D. licha*, *T. alalunga*, *K. pelamis* and *S. cretense* were less centralized and use more specialized approaches to feeding than those positioned towards the center in the trait space. The high biomass removal of these species can have far-reaching impacts on ecosystem structure and function [1, 75]. For example, overexploitation of mid and top predators like *K. pelamis* and *X. gladius* can lead to increases in the abundance of their prey species, potentially altering predator-prey dynamics and community composition [10, 75]. Similarly, the removal of herbivorous species like the parrotfish *S. cretense* can affect the structure and health of corals and benthic communities by disrupting grazing patterns and algal abundance [3, 6, 76].

Life history results suggest that most landed species were non-guarders, with sizes less than 100 cm, high fecundity, and low generation times. In contrast, species positioned on the opposite side of the trait space, such as *C. conger*, *X. gladius*, and the Elasmobranchii species *D. licha*, exhibited different life history traits. For instance, high fecundity and low generation time contribute to the reproductive potential and population growth rates of landed species, but they may also make them more vulnerable to overexploitation, as rapid population growth may lead to increased fishing pressure and depletion of stocks if not managed sustainably [77]. In this sense, Pinsky et al. [77] analyzed traits, including growth rate, fecundity, egg diameter and trophic level of 154 marine fish populations around the world in relation to fishing pressure and climate variability. Their study revealed that species with a faster growth rate were at a greater risk of collapse and exhibited lower relative population levels. These authors concluded that species with fast life histories are more susceptible to collapse and sensitive to overfishing. Additionally, species positioned on the opposite side of the trait space exhibiting contrasts in life history strategies, are likely more vulnerable to overexploitation and changes in ecosystem structure [8]. For example, species with low fecundity and long generation times, such as the deep-sea sharks *D. licha*, *Centrophorus squamosus*, and *Centroselachus crepidater*, may be particularly vulnerable to overexploitation, especially considering that most of the Azorean deep-sea sharks are caught as bycatch by bottom longline fishery [68, 78]. Their slower population growth rates, long longevity and lower reproductive output increase the risk of population decline and local extinctions [8, 79]. Life history traits of deep-sea sharks are very sensitive and crucial for their reproductive success and survival rates, ultimately impacting populations and regulating their dynamics. Consequently, this can have significant implications for ecosystem function and processes, influencing the food web and nutrient cycle in the deep-sea environment [8, 80].

## 4.3 Implications for fisheries management

These results hold significant ecological implications for the resilience and sustainability of exploited fish populations. Understanding the distribution of species within the functional space regarding habitat use function can provide valuable insights for fisheries management [11, 15, 20, 23]. For instance, identifying clusters of target species with similar habitat

preferences can assist managers in directing conservation efforts or implementing spatial management measures to safeguard critical ecological functions and habitats while monitoring fishing impacts [65]. By protecting habitats that support a diverse range of life-history strategies, managers can mitigate the impacts of fishing activities and enhance the adaptive capacity of marine ecosystems to environmental changes.

Seasonal fluctuations and increased catches of specific fish species can significantly deplete traits and functions from ecosystems, impacting their structure, dynamics, and resilience [1, 8, 71, 78]. Understanding and mitigating these ecological impacts involves considering metrics weighted by catch, such as FEve, FDiv, and FDis, to better manage seasonal fishing pressures [7, 17–20]. This includes targeting species with traits linked to high landings. Incorporating life history traits into sustainable management strategies is crucial for ensuring long-term fisheries viability. For species with high fecundity and short generation times, measures such as size limits, catch quotas, and seasonal closures are essential to prevent overexploitation. Species with different life history strategies, such as elasmobranchs, may require more cautious management, including stricter catch limits, protected areas, and measures to reduce bycatch. In the Azores, regulations like ordinance n˚ 21/2019 establish minimum catch sizes for species such as *H. dactylopterus*, *C. conger*, *P. bogaraveo*, and various Elasmobranchii species. Ordinance n˚ 92/2019 sets annual landing limits for many species targeted by local and coastal fleets, including those identified in our study with unique trait combinations (vertex species) such as *R. clavata* and *S. cretense*. This ordinance also prohibits the capture of Elasmobranchii species such as *Isurus* spp., *G. galeus*, and *P. glauca* in Azorean waters. Therefore, considering the intraspecific trait variability of exploited fish and their associated functions over time is essential for adaptive fisheries management, conservation efforts, and maintaining ecosystem functioning and services.

## 4.4 Study limitations and future research directions

It is important to highlight the limitations of the present study and identify key aspects for future research. The current study lacked information on unreported catches, which means the potential effects of fishing pressure on unreported species were not investigated. Future studies should incorporate data on unreported catches to provide a more comprehensive understanding of fishing impacts on ecological roles played by all species caught. Although the Azorean fleets primarily use selective fishing gear such as hooks and lines, it is essential to consider the effects of fishing pressure on fish assemblages in a multi-gear context. Different gear types can significantly affect trait diversity [15, 81]. Additionally, the study did not evaluate the predatory approaches of the species, including their vertical migrations during feeding activities. These behaviors can significantly impact ecosystem dynamics and species interactions, so future research should examine these predatory behaviors to understand their impacts on ecosystem functioning. Furthermore, it is crucial to assess the effects of environmental factors and climate change on the traits of reported and unreported species [82]. Such changes have been reported to significantly alter essential traits, such as size and age at first maturity and fecundity over time [83]. Finally, considering life-history traits as numeric rather than categorical – for instance, size and age at first maturity, asymptotic length, growth rate, longevity, fecundity, mortality rate, larval pelagic duration, and spawning season length – would allow for the use of a wide range of statistical methods, providing deeper insights into trait diversity, ecological roles and ecosystem functioning. This approach would also offer a better understanding of the evolution of species populations under fishing and climate-induced pressures.

## 5. Conclusions

The findings of the present study provide insights into the complex interplay between fishing activities, species traits, and ecosystem dynamics in the waters around the Azores. Despite the absence of significant changes in trait diversity over the past four decades, the landed species exhibit a wide trait diversity, particularly traits related to habitat use and life history. Seasonal availability and increased catch of specific fish species can result in the significant removal of traits and functions from the ecosystem, as indicated by the sensitivity of metrics weighted by landings. These findings underscore the need for adaptive management strategies that consider the selective pressures exerted by fishing activities on species or the diversity of traits and ecosystem functions. Such strategies are essential to ensure the long-term sustainability of fisheries and marine ecosystems.

## Supporting information

**S1 Table. List of fish species reported in the Azorean landings.**
(DOCX)

**S2 Table. Comparison of functional diversity metrics between local and coastal fishing for fish landed in the Azores archipelago.** nbsp: number of species, sing.sp: species functionally different in the landings, quali.FRic: quality of the reduced-space representation required to compute FRic and FDiv, FRic: functional richness, FEve: functional evenness, FDiv: functional divergence, FDis: functional dispersion. P-values represent significance between local and costal pair for each metric in each analysis from randomization testing, considering combined trait modalities (habitat use, locomotion, feeding and life history).
(DOCX)

**S3 Table. Pairwise comparison of functional diversity metrics across functional modalities and decades for species landed by local fishing in the Azores archipelago.** FRic: functional richness, FEve: functional evenness, FDiv: functional divergence, FDis: functional dispersion. Significant p-values (p<0.05) are highlighted in red, while marginally significant values are shown in orange. Differences refers to the differences between the observed and the mean values derived from the simulated distribution.
(DOCX)

**S4 Table. Pairwise comparison of functional diversity metrics across functional modalities and decades for species landed by coastal fishing in the Azores archipelago.** FRic: functional richness, FEve: functional evenness, FDiv: functional divergence, FDis: functional dispersion. Significant p-values (p<0.05) are highlighted in red, while marginally significant values are shown in orange. Differences refers to the differences between the observed and the mean values derived from the simulated distribution.
(DOCX)

**S5 Table. Pearson correlation ($r^2$) between the axes of the fuzzy correspondence analysis (FCA) and functional trait categories for habitat use modality.** Correlations higher than 0.50 and lower than -0.50 are highlighted in green and red, respectively.
(DOCX)

**S6 Table. Pearson correlation ($r^2$) between the axes of the fuzzy correspondence analysis (FCA) and functional trait categories for locomotion modality.** Correlations higher than 0.50 and lower than -0.50 are highlighted in green and red, respectively.
(DOCX)

**S7 Table. Pearson correlation ($r^2$) between the axes of the fuzzy correspondence analysis (FCA) and functional trait categories for feeding modality.** Correlations higher than 0.50 and lower than -0.50 are highlighted in green and red, respectively.
(DOCX)

**S8 Table. Pearson correlation ($r^2$) between the axes of the fuzzy correspondence analysis (FCA) and functional trait categories for life history modality.** Correlations higher than 0.50 and lower than -0.50 are highlighted in green and red, respectively.
(DOCX)

**S9 Table. List of species positioned at the vertices of the functional spaces, possessing the most unusual combination of traits.**
(DOCX)

**S1 Fig. Illustration of diet data transformation for four species, demonstrating the data transformation from categorical binary to fuzzy coding.**
(DOCX)

**S2 Fig. Exploring the relationships between functional traits and fish species within each function (habitat use, locomotion, feeding, and life history) in the functional space (areas in blue, green, light red, and light orange).** The plots of the fuzzy correspondence analysis (FCA), .i.e., A, C, E and G, illustrate the correlation between the FCA axes (axes 3 and 4), and functional trait modalities (See also S4–S7 Tables and S1 Fig). Different colors represent various trait modalities in FCA plots. In the plots B, D, F, and H, the fifteen most landed species and their total landings in tonnes are highlighted in red. Abbreviation codes available in **Table 1**. The scientific names of each vertex species highlighted in blue can be seen in **S9 Table**.
(DOCX)

## Acknowledgments

We would like to thank João Carlos Santos (OKEANOS-UAc) for the organization of the landing data of the Lotaçor/OKEANOS-UAc) database, and to Dr. Filipe M. Porteiro for assisting with the species names. We would like also to extend our sincere gratitude to the referees for their valuable contributions and insightful comments improving our article. [20, 29, 32]

## Author Contributions

**Conceptualization:** Eudriano F. S. Costa, Gui M. Menezes, Ana Colaço.

**Data curation:** Eudriano F. S. Costa.

**Formal analysis:** Eudriano F. S. Costa.

**Funding acquisition:** Gui M. Menezes, Ana Colaço.

**Investigation:** Eudriano F. S. Costa.

**Methodology:** Eudriano F. S. Costa.

**Validation:** Eudriano F. S. Costa.

**Visualization:** Eudriano F. S. Costa.

**Writing – original draft:** Eudriano F. S. Costa.

**Writing – review & editing:** Eudriano F. S. Costa, Gui M. Menezes, Ana Colaço.

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
