## [Decision Letter · Decision Letter 0]

4 Jul 2024

PONE-D-24-22486Trait diversity in exploited fish of the Azorean islands: Insights from a multifunctional approachPLOS ONE

Dear Dr. Costa,

Thank you for submitting your manuscript to PLOS ONE. After careful consideration, we feel that it has merit but does not fully meet PLOS ONE’s publication criteria as it currently stands. Therefore, we invite you to submit a revised version of the manuscript that addresses the points raised during the review process. The MS is well written, structured and very interesting, with a clear and reliable methodology. Data provided by the MS are essential to improve the knowledge base regarding the effect of anthropogenic pressure on teleost species and stocks. There are some major concerns that should be fixed, fllowing the suggestions and comments provided by the reviewers, in order to improve the MS quality, clarity and readibility.

We look forward to receiving your revised manuscript.

Kind regards,

Claudio D'Iglio, Ph.D.

Academic Editor

PLOS ONE

Journal Requirements:

   "This work was performed under the framework of the project FunAzores co-funded by AÇORES 2020, through the FEDER fund from the European Union: ACORES 01-0145-FEDER-000123. Okeanos team received national funds through the FCT – Foundation for Science and Technology, I.P., under the project UIDB/05634/2020 and UIDP/05634/2020 and through the Regional Government of the Azores through the initiative to support the Research Centers of the University of the Azores and through the project M1.1.A/REEQ.CIENTÍFICO UI&D/2021/010. AC is supported by the national funds through the FCT within the scope of CEECIND/ 00101/2021 and https://doi.org/10.54499/2021.00101.CEECIND/CP1669/CT0001"

   "The authors declare that the research was conducted in the absence of any commercial or financial relationships that could be construed as a potential conflict of interest."

5. We note that Figure 1 in your submission contain map/satellite images which may be copyrighted. All PLOS content is published under the Creative Commons Attribution License (CC BY 4.0), which means that the manuscript, images, and Supporting Information files will be freely available online, and any third party is permitted to access, download, copy, distribute, and use these materials in any way, even commercially, with proper attribution. For these reasons, we cannot publish previously copyrighted maps or satellite images created using proprietary data, such as Google software (Google Maps, Street View, and Earth). For more information, see our copyright guidelines: http://journals.plos.org/plosone/s/licenses-and-copyright.

Additional Editor Comments:

The MS of Costa et al. provide very interestinf and important data regarding the effect of fishery on the functional traits diversity of several marine fish species. These are essential information to understand the effect of anthropogenic pressure related to fishery, in order to improve the conservation of heavely exploited species and stocks.

The paper is clear, well structured with a well defined, clearly presented, methodology, but despite this there are some major concern regarding the clearity and readibility of the paper (especially regarding the Discussion and Introduction sections) to be fixed to improve th MS quality.

Reviewers' comments:

Reviewer's Responses to Questions

**Comments to the Author**

1. Is the manuscript technically sound, and do the data support the conclusions?

Reviewer #1: Partly

Reviewer #2: Partly

2. Has the statistical analysis been performed appropriately and rigorously? 

Reviewer #1: Yes

Reviewer #2: Yes

3. Have the authors made all data underlying the findings in their manuscript fully available?

Reviewer #1: No

Reviewer #2: Yes

4. Is the manuscript presented in an intelligible fashion and written in standard English?

Reviewer #1: No

Reviewer #2: Yes

5. Review Comments to the Author

Reviewer #1: Article: Trait diversity in exploited fish of the Azorean islands: Insights from a multifunctional approach

Manuscript ID: PONE-D-24-22486

General Comments

Costa and colleagued performed a study to detect the effect of fishery on the functional traits diversity of several marine fish species.

The paper is well structured and well written. The tested hypotheses are well defined, as the methods used too. Despite this, a major problem of the paper relies on the lengthy of both introduction and discussion that make the paper difficult to be followed in some parts.

Specific comments

Concerning all the manuscript:

Please carefully check all the scientific names and use the correct zoological nomenclature, each first time a species is mentioned, e.g. Pagellus bogaraveo (Brünnich, 1768). After the first time P. bogaraveo can be used. Apply this rule for all the mentioned species. “Sparissoma cretense” (line 355)

Title:

The title well frames only part of the performed study, and it should be modified to better reflects all the parts of the study. It would be better to highlights, starting from the title, “the potential impacts of exploitation on the ecological roles of fish species targeted by fisheries”. Moreover, it is no clear which traits are focused. Add “functional”?

Abstract:

Lines 44-47: which findings highlighted the importance of adaptive management strategies? Please briefly state here or re-arrange the abstract conclusion.

Keywords:

“ecosystem functioning” is a bit off topic. Please consider replacing with another keyword.

Introduction:

Despite the authors well introduce their study, as already motioned, some parts of the introduction are too lengthy bringing the lecturer to information far from the paper. All the paragraph related to Azores fisheries can be drastically reduced to few lines.

Material and methods:

2.2 “Lading data” � Landing

It is necessary to present the list of total fish species as part of this paper. It is unconceivable that a reader has to switch to another paper to see on which species the study is performed.

Table 1: Swimming mode � Correct the following: “Thunniform(Tetr)” (Tetr) is already used for Tetraodontiform.

Discussion:

Discussion section is very detailed, but it needs to be reduced, avoiding redundance of some sentences. This could help in improving readability of the paper and the focus on specific results obtained.

Conclusion:

I suggest the authors be more cautious in drawing the conclusions of their study.

As the authors state in different points of their paper there is, in the results “the absence of significant variations in trait diversity across decades…”; then, drawing strong conclusions without a historical data base is not possible. Please avoid in the conclusion the use of terms as “valuable”, it is a bit self-aggrandizing.

Reviewer #2: I found this manuscript interesting, well-written and with a huge amount of data correctly analyzed. As reported by the authors, it represents the first attempt to connect trait diversity to fish landings over time through a comprehensive analysis using null models, a nice approach which led to interesting results, even if limited in some cases by the strict relation among fishing activities and ecosystems dynamics, which, as the authors know, match with multiple aspects of the anthropogenic pressures, such as climate changes, pollution, habitat removal for coastal areas, etc. However, considering the novelty of the study for the area, and the good results obtained, even this strictly focused view could result important for future studies in this field, also to deepen other aspects starting from the data here reported by Eudriano F.S. Costa and colleagues.

Introduction section missed an important aspect of fisheries research, linked to the use of different fishing methods and their relapses in marine environment. Indeed, by-catch represents an important source of bias in scientific data collection, because often discarded after capture without landing. At the same time, essential species for marine ecosystems are often collected within this undetected resource, leading to importante consequences in habitats equilibria and their protection's management. Please add a period about this topic in this section, to contextualize better the treated topic also considering the important limitations which still today affect the landed's databases from a biological point of view.

Try to improve the quality of all the Figures, especially the ones that reports written informations, such as 2 and 5, which resulted very hard-to-read in the present form.

Table 1: please add more details in the caption about the trophic position and growth coefficient values.

The period among lines 206 and 217 could results more appropriate in introduction section, due its more general view compared to the subsequent part of the paragraph which on the contrary resulted focused on the methodology of present study.

Line: 319: Are the authors referring as "types" to local and coastal fishing vessels reported in lines 157-158? Please clarify and better argue this first essential sentence of the results section. Considering the amount of parameters and variables (correctly) analyzed in your study, being more clear as possible in focal points could results essential for the readers.

Within the results section, try to reduce comments more similar to discussion than results, such as lines 354, 366 (..indicating..), 382 (..suggest..). I understand is not simple to totally separate the concepts of results and discussion in this kind of manuscript (maybe more adapt to a style as "results & discussion" section), but try your best for manuscript fluency, if possible.

I understand the focus of this study, and agree with, but the main limitation of this document is represented by the climate changes effects occurred during the studied period all over the world, comprises the treated area, which were not mentioned at all. Indeed, even if fishing activities represents a strong source of alterations for marine organisms, a wider approach should be adopted when evaluating ecosystem equilibria, as in this case. At least mentioning as limitation of the study would be correct, highlighting the importance to cross multiple approaches to this problem. Indeed, as stated by the authors about the decline in ecosystem functions, and hypothesized from literature for the Azores (reported in line 424) "..particularly due to fishing activities", the reader is induced in thinking also to other untreated aspects, over fishing activities. Also, the period about life traits and reproductive cycle aspects (lines 593-619) couldn't ignore these additional aspects involved in environmental changes, which are reported to strongly modified essential functions, as for example: spawning, hatching and survival of teleosts.

Another important source of potential biases which needs to be better argued is represented by fishing regulation in the Azores Islands in the last four decades, interested by the study. For example, in lines 535-536 the authors reports: "Implementing fishery measures such as quotas, minimum landing size, area, and temporal fishing closures...(is crucial to mitigate the impact of fishing on ecosystem functions and processes in Azores)". Were these aspects uniform during the data collection's period? If not, what possible influences in the collection of analyzed data could there be? Hypothesis about the previous management's effects on these resources (and the analyzed data)?

Lines 555-576: in this context, it would be also important to evaluate the different predatory approaches of the studied species, which sometimes use also vertical migrations during their feeding activity, sometimes not. Moreover, also different fishing gears, baited or not, sometimes drives the species in moving during their feeding activity. I understand that these are not simple aspects to evaluate without specific approaches, but, at the same time, should be mentioned among the possible sources of biases.

Add the potential limitations of this study to the conclusion section.

Double-check references list for scientific name and style.

Best regards

The reviewer

6. PLOS authors have the option to publish the peer review history of their article (what does this mean?). If published, this will include your full peer review and any attached files.

Reviewer #1: No

Reviewer #2: **Yes: **Marco Albano

---

## [Author Response · Author response to Decision Letter 0]

18 Jul 2024

RESPONSES TO THE REVIEWER #1

General comments: “The paper is well structured and well written. The tested hypotheses are well defined, as the methods used too. Despite this, a major problem of the paper relies on the lengthy of both introduction and discussion that make the paper difficult to be followed in some parts.”

Response: 

Thank you for your positive feedback on the structure and clarity of our paper, as well as on the well-defined hypotheses and methods. We appreciate your constructive criticism regarding the length of the introduction and discussion sections. We reviewed these sections carefully in order to improve readability and coherence. However, we have inserted a new subhead in the Discussion according to reviewer’s #2 suggestions (Please see Lines 634-658).

Specific comments

Comment 1: “Concerning all the manuscript: Please carefully check all the scientific names and use the correct zoological nomenclature, each first time a species is mentioned, e.g. Pagellus bogaraveo (Brünnich, 1768). After the first time P. bogaraveo can be used. Apply this rule for all the mentioned species. “Sparissoma cretense” (line 355)”

Response: 

Line 464: “…Conger conger...” was changed to “…C. conger...”

Line 464: “…Pagellus bogaraveo...” was changed to “…P. bogaraveo...”

Line 465: “…Helicolenus dactylopterus...” was changed to “…H. dactylopterus...”

Line 465: “…Phycis phycis...” was changed to “…P. phycis...”

Line 485: “…Pagellus bogaraveo...” was changed to “…P. bogaraveo...”

Line 506: “…Pagellus bogaraveo...” was changed to “…P. bogaraveo...”

Line 506: “…Dalatias licha...” was changed to “…D. licha...”

Lines 525-526: “…Helicolenus dactylopterus...” was changed to “…H. dactylopterus...”

Line 549: “…Conger conger...” was changed to “…C. conger...”

Line 550: “…Raja clavata...” was changed to “…R. clavata...”

Line 570: “…Conger conger...” was changed to “…C. conger...”

Line 570: “…Xiphias gladius...” was changed to “…X. gladius...”

Lines 620-621: “…Helicolenus dactylopterus...” was changed to “…H. dactylopterus...”

Line 625: “…Galeorhinus galeus...” was changed to “…G. galeus...”

Line 626: “…Prionace glauca...” was changed to “…P. glauca...”

Comment 2: “Title: The title well frames only part of the performed study, and it should be modified to better reflects all the parts of the study. It would be better to highlights, starting from the title, “the potential impacts of exploitation on the ecological roles of fish species targeted by fisheries”. Moreover, it is no clear which traits are focused. Add “functional”?”

Response: 

Lines 1-3. Title: “Trait diversity in exploited fish of the Azorean islands: Insights from a multifunctional approach” was changed to “The potential impacts of exploitation on the ecological roles of fish species targeted by fisheries: a multifunctional perspective”

We believe that the new title better reflects our study, highlighting its relevance and scope, and making it more informative to the target audience. Additionally, the word “multifunctional” indicates that that the study examined various traits associated with different ecological roles, and how these traits may contribute to the overall functioning of the ecosystem.

Comment 3: “Abstract: Lines 44-47: which findings highlighted the importance of adaptive management strategies? Please briefly state here or re-arrange the abstract conclusion. Keywords: “ecosystem functioning” is a bit off topic. Please consider replacing with another keyword.”

Response: 

Line 45-47: “The findings highlight the importance of adaptive management strategies to address fishing impacts on species traits and their ecological roles, crucial for long-term fisheries and ecological sustainability.” was changed to “The findings highlight the importance of addressing fishing impacts on species traits and their ecological roles, which is crucial for long-term fisheries and ecological sustainability.”

Keywords: “ecosystem functioning” was changed to “ecological roles”.

Comment 4: “Introduction: Despite the authors well introduce their study, as already motioned, some parts of the introduction are too lengthy bringing the lecturer to information far from the paper. All the paragraph related to Azores fisheries can be drastically reduced to few lines.”

Response: 

The indicted paragraphs were shortened as following below:

Lines 89-101: “Fisheries play a vital role in the Azores Archipelago. Despite its rich marine biodiversity, fishing grounds are rare, small, and scattered, with less than 1% of the Exclusive Economic Zone (EEZ) having depths less than 600 m, and an average depth of 3000 m [24,25]. The Azorean fishing industry has expanded since the 1980s from the island shelves to offshore seamount areas and deeper waters. This expansion was facilitated by advancements in air transportation of fresh fish products to mainland Portugal and foreign countries, as well as by the adoption of boats equipped with high-tech equipment such as sonar, which increased the autonomy of fishing vessels [26–28]. Azorean fisheries are classified as small-scale because approximately 60% of the vessels are under nine meters in length [28]. Small scale fishing represents between 80-90% of the Azorean fishing fleet [26,27,29]. This fleet targets a variety of fish species with different economic values, playing different ecological roles at various depths in nearshore and seamounts areas [20,24,28,30,31]. The most important targets are tuna and tuna-like species (in weight), deep-water demersal species (in value), and small pelagic species [28]. These species are landed at Azores Auction services for commercialization by two types of fishing fleets: local and coastal fleets. The former comprises small boats less than 12 meters in length, operating in proximity to coastal areas at depths up to 700 m with limited autonomy, while the latter explores more distant waters deeper than 700 m, focusing on large pelagic and deep-water demersal species [27,32]. Both fleets utilize high selective gears employing hook and line, including a variety of handlines and longlines, which are the most important fishing methods. However, traps and small purse-seine methods may be used by both fleets, but they are more commonly utilized by local fishing operations [27,28].” was changed to “Fisheries are crucial in the Azores Archipelago, which, despite its rich marine biodiversity, has limited fishing grounds with less than 1% of its EEZ being shallower than 600 m and an average depth of 3000 m [24,25]. Since the 1980s, the Azorean fishing industry has expanded from island shelves to offshore seamount areas and deeper waters, driven by advancements in air transport of fresh fish and high-tech equipment on boats, such as sonar [26–28]. Classified as small-scale, about 60% of Azorean vessels are under nine meters in length, and small-scale fishing comprises 80-90% of the fleet [26,27,28,29]. This fleet targets various species, including tuna, deep-water demersal, and small pelagic species, across different depths [20,24,28,30,31]. Fish are commercialized through local and coastal fleets: the local fleet operates small boats (under 12 meters) near coastal areas up to 700 m deep, while the coastal fleet targets deeper waters over 700 m [27,32]. Both fleets primarily use selective hook and line gears, such as handlines and longlines, with traps and small purse-seine methods also used, especially by local operations [27,28].”

Lines 102-110: “The small-scale fisheries in the Azores are recognized for their sustainable practices, supported by an efficient and unique system for fishery data collection established since the 1970s, and local regulation measures [33,34]. These regulations include area and temporal fishing closures (e.g., local regulation ordinance n° 74/2015), species-specific quotas, and minimum landing sizes (e.g., local regulation ordinance n° 21/2019). Furthermore, there are limits for annual landings of many Actinopterygii species, including Phycis phycis, Helicolenus dactylopterus, Serranus atricauda, and Pagellus bogaraveo, and , and the forbitten to target two shark species Galeorhinus galeus and Prionace glauca (e.g., local regulation ordinances n° 92/2019 and n° 27/2023). Despite all efforts to manage and promote the sustainable use of the main fishery resources in Azorean waters, there is still a lack of knowledge regarding the potential impacts of fishing activities on marine biodiversity, ecosystem function, and services.” was changed to “The small-scale fisheries in the Azores are recognized for their sustainable practices, supported by an efficient fishery data collection system established since the 1970s and local regulations [33,34]. These regulations include fishing closures (e.g., ordinance n° 74/2015), species-specific quotas, and minimum landing sizes (e.g., ordinance n° 21/2019). Limits are set for annual landings of species like Phycis phycis, Helicolenus dactylopterus, Serranus atricauda, and Pagellus bogaraveo, with a ban on targeting sharks Galeorhinus galeus and Prionace glauca (e.g., ordinances n° 92/2019 and n° 27/2023). Despite efforts for sustainable resource use, knowledge gaps remain regarding the potential impacts of fishing on marine biodiversity and ecosystem functions.”

Comment 5: “Material and methods: 2.2 “Lading data” � Landing It is necessary to present the list of total fish species as part of this paper. It is unconceivable that a reader has to switch to another paper to see on which species the study is performed. Table 1: Swimming mode � Correct the following: “Thunniform (Tetr)” (Tetr) is already used for Tetraodontiform.”

Response: 

Line 46: “The list of all fish species landed in Azores, along with the total landings by species, can be found in Costa et al. [20], and Costa et al. [29].” was changed to “All fish species included in this study are listed in S1 Table. For more information about all fish species reported in the Azorean landings, as well as the total landings by species, can be found in Costa et al. [20], and Costa et al. [29].”

Supplementary information: “S1 Table” was inserted.

Supplementary information: “S1 Table” was changed to “S2 Table”.

Supplementary information: “S2 Table” was changed to “S3 Table”.

Supplementary information: “S3 Table” was changed to “S4 Table”.

Supplementary information: “S4 Table” was changed to “S5 Table”.

Supplementary information: “S6 Table” was changed to “S7 Table”.

Supplementary information: “S7 Table” was changed to “S8 Table”.

Supplementary information: “S8 Table” was changed to “S9 Table”.

Line 308: “…from 22.5 % to 30.5 % (S1 Table).” was changed to “…between the two fishing types (S2 Table).”

Line 336: “…from 22.5 % to 30.5 % (S4-S7 Tables).” was changed to “…from 22.5 % to 30.5 % (S5-S8 Tables).”

Line 348: “…can be found in S4 Table.” was changed to “…can be found in S5 Table.”

Line 361: “…FCA locomotion can be found in S5 Table.” was changed to “…FCA locomotion can be found in S6 Table.”

Line 376: “…FCA feeding can be found in S6 Table.” was changed to “…FCA feeding can be found in S7 Table.”

Line 392: “…FCA life history can be found in S7 Table.” was changed to “…FCA life history can be found in S8 Table.”

Line 407: “…S2 Table and Costa et al. (2024a)” was changed to “…S3 Table and Costa et al. (2024a)”.

Lines 415-416: “…For details see S3 Table and Costa et al. (2024a).” was changed to “…For details see S4 Table and Costa et al. (2024a).”

Lines 422-423: “…and functional trait modalities (See also S4-7 Tables and S1 Fig).” was changed to “…and functional trait modalities (See also S5-S8 Tables and S1 Fig).”

Line 426: “…highlighted in blue can be seen in S8 Table.” was changed to “…highlighted in blue can be seen in S9 Table.”

Lines 183-184 - Table 1: “Thunniform (Tetr)” was changed to “Thunniform (Thun)”.

Comment 6: “Discussion: Discussion section is very detailed, but it needs to be reduced, avoiding redundance of some sentences. This could help in improving readability of the paper and the focus on specific results obtained.”

Response: 

Thank you for your feedback. While the reviewer did not specify which paragraphs to shorten or where to reduce redundancy. Nevertheless, we have made adjustments to enhance clarity and conciseness throughout the text. These changes were aimed at maintaining focus and clarity while preserving depth of content. Regarding the topic “4.2 Fuzzy Correspondence Analysis”, we have decided not to shorten this section because we consider its content essential for understanding the potential impacts of fisheries on traits relative to their functions. This analysis provides crucial insights the contribute significantly to the overall comprehension of our study’s findings.

Lines 442-454: “Fishing activities have the potential to influence diversity metrics estimation and alter the composition of functional traits within ecosystems over time by modifying both the species composition of catches and the amount of biomass landed [7,15,17,20,71]. Indeed, Costa et al. [20] reported significant variations in FRic, and functional space of fish species landed in the Azores for the last three decades. These authors attributed such changes to variations of species that possessed an unusual combination of traits reported in annual landings. Additionally, Costa et al. [36] found significant changes in species composition landed by both local and coastal fishing in the Azores, highlighting the implications for diversity metric estimation such as functional redundancy (FRed) and functional vulnerability (FVul). However, neither of these studies conducted a comprehensive investigation into trait diversity as performed in the present study using null models. Thus, the present study fills this gap by revealing the absence of significant in FRic between fishing types and over time across functions, suggesting that azorean fisheries has targeted species with the same range of traits through both local and coastal fishing over the past four decades.” was changed to “Fishing activities can influence diversity metrics and alter functional traits within ecosystems by changing species composition and biomass landed [7,15,17,20,71]. Costa et al. [20] reported significant variations in FRic and functional space of fish species landed in the Azores over the past three decades, attributed to variations in species with unusual trait combinations in annual landings. Costa et al. [36] also found significant changes in species composition landed by both local and coastal fishing in the Azores, affecting diversity metrics such as functional redundancy (FRed) and functional vulnerability (FVul). However, these studies did not comprehensively investigate trait diversity. The present study fills this gap using null models, revealing no significant differences in FRic between fishing types and over time across functions, suggesting that Azorean fisheries have targeted species with a consistent range of traits through both local and coastal fishing over the past four decades.”.

Lines 455-473: “Despite the absence of significant variations in trait diversity across decades, it is important to highlight that species landed by Azorean fishing fleets exhibit a wide range of combination of traits, suggesting that the landed species occupy a diverse array of ecological niches. This holds true even when considering only the fifteen most landed species. In azorean waters, fish are primarily caught using many different types of handlines and longlines. However, small pelagic species such as Trachurus picturatus, Scomber japonicus, and Boops boops are commonly caught using purse-seine nets [27,28,72]. Mbaru et al. [15] has suggested that the selective gear types used target a narrower range of functional diversity, implying that the targeted fish species may not exhibit a high diversity of traits, and potentially impact fewer ecosystem functions. In the Azores, the majority of selective gear types are associated with bottom habitats to capture species such as Raja clavata (Elasmobranchii), P. bogaraveo, C. conger, H. dactylopterus, Mora moro, P. phycis, Beryx spp., among others [72,73]. Menezes et al. [24] reported that the species composition of fish assemblages in the Azores were closely related to 

---

## [Decision Letter · Decision Letter 1]

29 Jul 2024

The potential impacts of exploitation on the ecological roles of fish species targeted by fisheries: a multifunctional perspective

PONE-D-24-22486R1

Dear Dr. Costa,

We’re pleased to inform you that your manuscript has been judged scientifically suitable for publication and will be formally accepted for publication once it meets all outstanding technical requirements.

Kind regards,

Claudio D'Iglio, Ph.D.

Academic Editor

PLOS ONE

Reviewers' comments:

Reviewer's Responses to Questions

**Comments to the Author**

1. If the authors have adequately addressed your comments raised in a previous round of review and you feel that this manuscript is now acceptable for publication, you may indicate that here to bypass the “Comments to the Author” section, enter your conflict of interest statement in the “Confidential to Editor” section, and submit your "Accept" recommendation.

Reviewer #1: All comments have been addressed

Reviewer #2: All comments have been addressed

2. Is the manuscript technically sound, and do the data support the conclusions?

Reviewer #1: Yes

Reviewer #2: Yes

3. Has the statistical analysis been performed appropriately and rigorously? 

Reviewer #1: Yes

Reviewer #2: Yes

4. Have the authors made all data underlying the findings in their manuscript fully available?

Reviewer #1: Yes

Reviewer #2: Yes

5. Is the manuscript presented in an intelligible fashion and written in standard English?

Reviewer #1: Yes

Reviewer #2: Yes

6. Review Comments to the Author

Reviewer #1: Dear Authors,

thank you so much for considering all my comments and for addressing relative revisions.

All the best regards

Reviewer #2: Dear Authors,

thank your to seriously revised your document considering all my previous comments. The manuscript looks more clear and complete now in some parts and your argumentation for hard-to-solve comments were reasonable.

Best regards

7. PLOS authors have the option to publish the peer review history of their article (what does this mean?). If published, this will include your full peer review and any attached files.

Reviewer #1: **Yes: **Gioele Capillo

Reviewer #2: **Yes: **Marco Albano

---

## [Editor Report · Acceptance letter]

2 Aug 2024

PONE-D-24-22486R1 

PLOS ONE

Dear Dr. Costa, 

I'm pleased to inform you that your manuscript has been deemed suitable for publication in PLOS ONE. Congratulations! Your manuscript is now being handed over to our production team.

Kind regards, 

on behalf of

Dr. Claudio D'Iglio 

Academic Editor

PLOS ONE